# Bond Modification of Carbon Rovings through Profiling

**DOI:** 10.3390/ma15165581

**Published:** 2022-08-14

**Authors:** Paul Penzel, Maximilian May, Lars Hahn, Silke Scheerer, Harald Michler, Marko Butler, Martin Waldmann, Manfred Curbach, Chokri Cherif, Viktor Mechtcherine

**Affiliations:** 1Institute of Textile Machinery and High Performance Material Technology (ITM), Technische Universität Dresden, 01062 Dresden, Germany; 2CARBOCON GMBH, 01067 Dresden, Germany; 3Institute of Concrete Structures (IMB), Technische Universität Dresden, 01062 Dresden, Germany; 4Institute of Construction Materials (IfB), Technische Universität Dresden, 01062 Dresden, Germany

**Keywords:** carbon reinforced concrete, bond behavior, bond mechanisms, profiling technology, tensile test, bond test

## Abstract

The load-bearing behavior and the performance of composites depends largely on the bond between the individual components. In reinforced concrete construction, the bond mechanisms are very well researched. In the case of carbon and textile reinforced concrete, however, there is still a need for research, especially since there is a greater number of influencing parameters. Depending on the type of fiber, yarn processing, impregnation, geometry, or concrete, the proportion of adhesive, frictional, and shear bond in the total bond resistance varies. In defined profiling of yarns, we see the possibility to increase the share of the shear bond (form fit) compared to yarns with a relatively smooth surface and, through this, to reliably control the bond resistance. In order to investigate the influence of profiling on the bond and tensile behavior, yarns with various profile characteristics as well as different impregnation and consolidation parameters are studied. A newly developed profiling technique is used for creating a defined tetrahedral profile. In the article, we present this approach and the first results from tensile and bond tests as well as micrographic analysis with profiled yarns. The study shows that bond properties of profiled yarns are superior to conventional yarns without profile, and a defined bond modification through variation of the profile geometry as well as the impregnation and consolidation parameters is possible.

## 1. Introduction

The use of fiber reinforcements in concrete has been established as an alternative to reinforcing steel in recent years. Research into the properties of TRC/CRC (textile/carbon reinforced concrete), TRM (textile reinforced mortar), or FRCM (fiber-reinforced cementitious matrices)—all synonyms for a composite made of continuous fiber-based reinforcement and a mineral-based matrix—is being conducted worldwide. Hereby, the studies focus primarily on strengthening concrete structures and fiber-reinforced composites as a high-performance material in civil engineering [1,2,3,4,5,6]. Impressions of the application potential in new constructions and building strengthening are, for example, filigree and precast concrete structures for bridges, façade panels, beams, shells, and pavilions, as well as strengthening of existing structures, especially bridges for shear and bending [7,8,9,10,11,12,13].

Continuous carbon fibers, also called rovings or Carbon Fiber Heavy Tows (CFHTs), are processed into grid-like structures or rods for use in concrete. Common carbon fiber rebars have diameters between 6 and 10 mm, thus in the size range of thin steel reinforcement rebars. They are preferably used in new building components. The carbon rovings that are processed into grid-like non-crimp fabrics (NCF) have significantly smaller diameters in the range of 1 to 3 mm. This makes them particularly suitable for use in filigree components such as façade panels or subsequent component reinforcement.

A capable bond between reinforcement and concrete is essential for load transfer and efficient utilization of the composite’s components. This requires knowledge of the acting bond mechanisms for continuous-fiber-based reinforcements and their interaction with the surrounding concrete.

Rovings for concrete constructions consist of several thousand individual filaments, e.g., [3,14]. In combination with the solvent spinning of the fibers, the rovings are given a sizing to facilitate handling and reduce sensitivity to mechanical damage in the textile manufacturing process. After the textile processing of the rovings to grid-like structures, they are provided with impregnation in an online or offline process. At this moment, the impregnation and consolidation of the roving (a) create the bond between the filaments in the yarn (inner bond) and thus ensure that all filaments participate equally in the load transfer as far as possible [15,16], and (b) influence the bond between the roving and the surrounding concrete matrix (outer bond).

The bond mechanisms in TRC/CRC are the subject of extensive research, e.g., [17,18,19,20]. It was recognized relatively early on that an effective internal bond, which is achieved by an even impregnation and consolidation of the roving, is a prerequisite for effective yarn utilization, as otherwise, the edge filaments are subjected to significantly greater stress than the core filaments, resulting in premature failure, e.g., [17,21]. Today’s common impregnations are generally based on styrene–butadiene rubber (SBR), epoxy resin (EP), or polyacrylate (PA). Each impregnation has a different effect on the internal and external bond but also the structural stability. SBR-impregnated reinforcements are usually more flexible and therefore suitable for curved shapes and reinforcements in existing structures (subsequent application), whereas stiff EP fabrics are preferred for precast slabs and plates. PA-impregnated grids with thermoplastic properties can be reshaped after textile processing and used for rectangular reinforcement structures. 

The bond between rovings and concrete is similar to steel reinforced concrete, mainly based on three mechanisms: adhesion, mechanical interlock, and friction (e.g., [18,22,23,24]). Adhesion results from the ingrowth of hydration products into the impregnation layer and/or fiber strand [14], adhesion or chemical bond between impregnation and cement paste. A relative displacement between roving and concrete destroys adhesion and activates the frictional resistance, which depends primarily on the roughness at the interface fiber strand and concrete matrix. It is adjustable by material abrasion ([21,23]). The form fit (also or shear bond) is the most important bond component in ribbed reinforcing steels. However, so far, form fit is relatively low with TRC/CRC. Contributions to the mechanical interlocking can be a (periodic) widening of the yarns between the crossing points in a grid-like NCF, the yarn waviness due to the roving constriction with the knitting thread, or cross yarns firmly connected to the yarn in the main carrying direction (see especially [20,23,25]). 

Due to the continuously ongoing development of raw materials, yarn processing techniques, impregnations, and concretes on the one hand and due to a large number of possible combinations of fiber reinforcements and concretes on the other, the bond mechanisms cannot be described in general terms to date, let alone allow a precise prediction and furthermore, a specific controlling of the bond behavior. However, important factors influencing the bond, such as the type and material characteristics of fiber and impregnation, the quantity of yarn or fabric geometry, textile binding and processing, concrete properties, and test conditions, such as temperature, are known, e.g., [20,22,23,24,26]. Accordingly, the proportion of the three relevant bond mechanisms in the overall bond resistance can vary. For textiles with low-modulus“soft”impregnation (e.g., SBR), the adhesive and frictional bond are decisive; for “stiff” textiles (high-modulus impregnation material such as EP or acrylate), the form-fit dominates [23,24]. In the case of conventional grid-like NCF, the form-fit effect is relatively small; therefore, the transmittable bond forces are comparatively low. Hence, the necessary reinforcement area is increased, which is further amplified by the imprecise predictable bond behavior resulting in additional over-dimensioning and high reduction factors for the design of the CRC. Therefore, the highly efficient use of the reinforcement structures is hindered.

Another difficulty in determining the bond resistance of CRC is, as yet, that one test setup has not been standardized. Single- or double-sided textile pull-out tests (e.g., [18,22,23]), overlap [22], and fiber strand pull-out tests [27] are used in various designs. From the tests, bond flow–crack opening, respectively, bond stress–slip relationships are derived (e.g., [18,25]). During all mentioned bond tests, different failures can occur. If the fiber strand breaks, the yarn tensile force is completely transferred into the matrix. In the case of yarn pull-out (interface failure), the bond length is too small—insufficient internal bond in the yarn results in telescopic yarn pull-out. In addition, the surrounding concrete can fail; in the case of newer generation textiles with higher yarn cross-sections and stiffer impregnation especially, there is a risk of splitting of the specimen (longitudinal cracking and/or splitting of the concrete cover), e.g., [20,22,23,25]. 

In summary, the bond behavior of continuous fiber-based reinforcement in concrete is not a trivial problem. For the application-specific design and construction of components, as well as strengthening layers made of CRC, a high-performance bond (high pull-out loads and bond stiffness) is a basic prerequisite. For increased material efficiency as well as suitability for use, a specific controllable composite would also be highly desirable in order to guarantee a long-lasting and predictable load-bearing capacity. According to our thesis, a defined profiling process of the yarns/rovings, including defined profile geometry, impregnation, and consolidation parameters, can specifically influence and control the bond resistance and, beyond that, the crack formation under load. Therefore, different profiles of carbon fiber rebars have already been studied in depth, and it has been shown that different geometries and varying rebar compositions determine the effectiveness of the TRC-composite [28,29]. This knowledge is now to be transferred to yarns because, in contrast to rods and rebars, yarns are flexible enough for further textile processing into windable, grid-like structures with better handling and higher productivity. In the paper, a promising approach for bond modification through targeted yarn profiling with adjustable profile, impregnation, and consolidation parameters, in addition to the first results of yarn analysis, as well as tensile and composite tests, are presented. 

## 2. Yarn Profiling Technology 

Different profiled rovings with varied impregnation and consolidation parameters were investigated for the development of CRC structures with controllable and predictable bond behavior on the basis of a defined form fit effect. In order to create profiled rovings with enhanced but also defined bond performance through a mechanical interlock with the concrete matrix yet maintaining high tensile properties and enough flexibility to be windable, a new profiling technology and innovative roving geometry were developed and patented at the Institute of Textile Machinery and High Performance Material Technology (ITM) [30]. The general process of the profiling method is shown in Figure 1. The characteristic of the roving geometry is the alternating, rectangular profile dents in the vertical and horizontal plane (Figure 2a,c), creating a so-called tetrahedral shape. 

To verify the qualities of the tetrahedral profiled rovings, the first tests were performed at ITM [31,32]. For the profiling, single Carbon Fiber Heavy-Tows (CFHTs) were impregnated in an aqueous polymer dispersion on an acrylate basis and placed in a developed profiling unit. The prototype unit consisted of profile bars that interlocked and formed a cavity when it was closed. The carbon roving acquired its profile as a negative form of the profiling unit. The profiling was permanently stabilized through the consolidation process of the polymer matrix under infrared (IR) radiation (see Figure 1).

During the profiling process, the CFHT acquired innovative and patented geometry in the form of a tetrahedral shape (Figure 2a). The profile is hereby defined by spacings between the dents in the same plane (horizontal or vertical) (here: 20 mm), the angular deviation from the linear orientation on the roving surface α (Figure 2b), and the difference between the minimum and maximum diameter in a dent (d_min_ and d_max_, resp.; Figure 2c).

In contrast to other shaping processes (helix, spiral, twisted, braided, etc.), the tetrahedral geometry distinguishes itself through a uniform reorientation of all filaments. Due to the alternation of the rectangular profile dents in the horizontal and vertical plane, the filaments are reoriented in such a way that all have the same deviation and, therefore, the same length between neighboring profile dents (Figure 3). Hence, all filaments distribute the load under strain in the same way: evenly and maintain their high tensile properties.

Through the developed manufacturing process, a very good yarn impregnation with the impregnation agent and a very dense filament arrangement in comparison to conventional manufacturing methods, such as a combination of multiaxial warp knitting techniques with online or offline impregnation processes, is predicted (see also Section 4.2). The thesis is that this clearly increases material utilization of the rovings (see Section 4.3) because an improvement of the inner bond causes a more even roving activation.

In order to enable a continuous and productive profiling process with high reproducible quality, a laboratory unit was developed and built at ITM [32,33,34] (see Figure 4a). 

In contrast to the prototype unit, it allowed continuous and endless production of the tetrahedral-shaped rovings. In principle, the laboratory unit worked according to the same process shown in Figure 1. The profiling was realized by an upper and lower circumferential chain with profiling tools that interlock when they meet. The yarn shape was limited and adjustable by the vertical distance of the two chains. During drying, cross-linking, and stabilization, the roving was clamped between the profiling tools of the chain (see Figure 4b).

## 3. Materials, Test Program, and Testing Methods

### 3.1. Rovings with Different Configurations for Carbon Fiber Reinforcement

A carbon fiber heavy tow (CFHT) Teijin Tenax-E STS 40 F13 48K 3200 tex (Teijin Carbon Europe GmbH, Wuppertal, Germany) was selected to investigate the influence of profiling on the mechanical properties. All different rovings in this study were produced with this CFHT. Table 1 shows the properties of the dry yarn. The tensile strength was determined in single yarn tensile tests at ITM, according to ISO 3341 (see Section 3.3 and Section 4.2, [35,36]). 

For the impregnation, profiling and consolidation of the rovings two different impregnation agents called TECOSIT CC 1000 (CHT Germany GmbH, Tübingen, Germany) and Lefasol BT 91001-1 (Lefatex Chemie GmbH, Brüggen, Germany), which are both polymeric dispersions with a solid content of ca. 50% were used. The exact product properties are listed in Table 2. The difference between the two impregnation agents could have a minor effect on the test results (variation of tensile properties) and will be validated in further studies; the test setup will remain the same. 

In addition to the non-impregnated, straight rovings (dry yarn from a spool), impregnated, non-profiled rovings, and two profiled roving variants (medium and strong profile, see Table 3) were manufactured (impregnation and consolidation according to the general process in Figure 1 without profiling) and subsequently analyzed. Hereby, each variant was combined with the two different impregnation agents Tecosit and Lefasol. As a reference (short “Ref.”), a single, straight roving extracted from the textile SITgrid 040 (Wilhelm Kneitz Solutions in Textile GmbH, Hof, Germany) with the same fiber material and impregnation agent Tecosit is used. The textile was tested during the project Carbon Concrete Composite (C^3^) [40] and represents a reliable reference. Table 3 shows the basic properties of these rovings.

For the investigation of the influence of the profiling process on the roving properties, tetrahedral profiled rovings were produced on the discontinuously working prototype unit (Series index “P”) as well as on the continuously working laboratory unit (see Table 4). The function of the prototype unit is described in Section 2 and is distinguished from the laboratory unit through a static shaping process and constantly applied pressure during the consolidation. The laboratory unit produced profiled rovings with different profile configurations (see Table 3).

The profile characteristics of the different rovings are shown in Table 3. Hereby, the profile of the tetrahedral-shaped rovings was characterized by the difference between the minimum and maximum diameter in a profile dent (smallest cross-section) and the angle of the filament orientation. According to Figure 2, the angle α was hereby determined as the tangent between the distance between two neighboring profile dents in the vertical and horizontal plane (10 mm) and the difference between the minimum and maximum diameter. The impregnated roving with no profile showed a circular shape with a diameter of about 2 mm, and the single roving extracted from the textile showed an elliptical cross-section due to the warp knitting process and the fixation with the knitting thread. 

In order to investigate the influence of the solid content of the impregnation as well as the consolidation parameters on the bond behavior (specific pull-out load and bond stiffness) of tetrahedral profiled rovings, test specimens with the same profile characteristics (strong profile—Series 4) were produced on the laboratory unit with different solid contents (30%, 40%, 50%) (see Table 4). To investigate if an intensified consolidation of the profiled rovings has an influence on the bond behavior, the series with the highest solid content (50%) and, therefore, presumably the highest bond performance was consolidated for 4 min as well as for 10 min (see Table 4). Hereby, the consolidation time varied through different production speeds of the continuously working Profiling unit. For the consolidation (drying and stabilization), the impregnated roving was positioned between several opposites positioned IR-modules from OPTRON GmbH (Germany) Typ IRDS750 SM 3kW (400 V) fast middle wave with 90% power (2.7 kW) and a distance of 50 mm to the roving. The solid content of the impregnation varied by adding water to the polymeric dispersion. 

### 3.2. Concrete Matrix

Fiber-based reinforcements are very often embedded in cementitious matrices with small maximum grain sizes (e.g., [14]). For such fine concrete matrices, the compressive strength and the flexural tensile strength are usually determined according to DIN EN 196-1 [41] after 28 days. Three standard prisms with a cross-section of 40 × 40 mm and a length of 160 mm were concreted per batch. First, the bending tensile strength was evaluated in a three-point bending test [41]. The compressive strength was then determined in a uniaxial compression test on the resulting two prism halves.

In the course of the initial trials on profiled yarns presented here, two different fine concretes were used. One was the TF 10 CARBOrefit^®^ (PAGEL Spezial-Beton GmbH & Co. KG, Essen, Germany) fine concrete. This cement-based, fine concrete dry mix has been established for the subsequent strengthening of structures in Germany over the last years [42,43]. The maximum grain size of the mixture is 1 mm. Only water needs to be added to the ready-mix. The soft plastic consistency is suitable for laminating in layers and for spraying. The factory-guaranteed properties are summarized in Table 5. These minimum values were met in all test series. The second mixture—a fine concrete-dry-mix (called BMK 45-220-2, consisting of binder material BMK-D5-1 from Dyckerhoff, Germany; KSM Compact III by KSM-Babst GmbH, Germany; fine sand BCS 412 from Strobel, Germany; sand 0/2 from Ottendorf, Germany; superplasticizer PCE SP VP-16-0205-02 from MC-Bauchemie, Germany and water)—was used for the pull-out tests at the Institute of Construction Materials (IfB) of TU, Dresden. The concrete properties were determined on 40 × 40 × 160 mm prisms according to DIN EN 196-1 [41]; the mean values are also listed in Table 5.

### 3.3. Test Program and Test Setups 

In order to visualize the impregnation quality of the yarns and to analyze the influence of the shaping process on the filaments, micrographic analyses were carried out on different roving sections (cross- and longitudinal sections). Hereby, EP-resinated roving samples were examined with a reflected-light microscope (Zeiss AxioImager.M1m from the Carl Zeiss AG, Jena, Germany) with a bright field and magnification factor of 200.

Tensile tests on single impregnated rovings are less time-consuming and a fast method to obtain a statement on the change in load-bearing capacity as a result of further processing, e.g., profiling. They are suitable for production control and characterization of the influence of various production parameters. The tests were conducted on the basis of DIN EN ISO 10618 [44] (see also [35,45]. The free yarn length was 200 mm. The ends of the profiled yarns were clamped with metal clamps with a steel file cut. For this purpose, the single impregnated rovings (without profile, with profile, and from textile) were resinated in the clamping area (Section 3.4) and clamped between two pneumatic pressured steel clamps (50 × 60 mm) with a file cut surface at 35 bar. Figure 5 shows the principle of the clamping on the left and the test facility on the right. All tests were performed with the testing machine Zwick 100 from ZwickRoell GmbH & Co. KG (Germany). The test speed was 3 mm/min. The entered force was measured with a 100 kN force tensor, and the elongation of the roving was determined with an optical laser system consisting of two length variation sensors and reflex markers, which were fixed on the roving prior to the test. The modulus of elasticity was calculated from the applied force during a roving elongation from 0.15 to 0.9%.

All tests on the CRC specimens took place at 20 °C, 28 days after casting. For the tensile test on textile reinforced concrete, usually, fabric sections are embedded in fine-grained concrete. In addition to the tensile strength, the cracking behavior of the composite can be analyzed. The specimen dimensions essentially depend on the grid-like non-crimp fabric geometry (specimen width), the yarn thicknesses (specimen thickness), and the fabric’s load-bearing capacity. Detailed recommendations can be found in [23,46]. The testing was conducted according to these suggestions. However, individual yarns which were embedded parallel and as stretched as possible in the concrete were tested. The test setup is shown in Figure 6. The test specimen was clamped into the testing machine at both ends. The specimen length included sufficiently long anchorage areas and a centric measuring section of at least 200 mm, which was not influenced by the lateral pressure in the load introduction area. The displacement transducer (DD1) was highly visible, clamped to the specimen in the middle area, and used to record the strain in the free measuring length.

There are various possibilities for the characterization of the bond between textile reinforcement and concrete; however, there is no standardized test method yet. 

Therefore, a test method suitable for single yarn pull-out tests was examined, mainly to understand whether this test method is suitable for the investigation of profiled yarns.

Single yarn pull-out (YPO) tests were conducted at the IfB in order to analyze the characteristic bond–slip behavior of single rovings with different profile properties (e.g., [27,47,48]). In this type of experiment, individual rovings were embedded in cubic concrete blocks. The upper block provided an embedment length of 50 mm at the top roving section. The lower block possessed an increased embedment length of 90 mm at the bottom roving section for a defined roving fixation. The concrete cover was 40 mm. The specimens were fixed in an upper and lower specimen holder, and the pull-out force—slip–deformation curve was measured by a single-sided pull-out in the upper concrete block with a controlled quasi-static load (Figure 7). The pull-out (slip) deformation was measured by an optical system consisting of laser sensors and aluminum clips, which were fixed to the yarn. 

### 3.4. Specimens Manufacturing

Short roving sections of about 10 mm were placed in cylinders of 20 mm diameter and fully resinated for the microscopic examinations. After one day of drying, the front side was ground with sandpaper and finally polished.

450 mm long roving sections were cut to size for the yarn tensile tests. Then, the rovings were stretched and clamped in a frame. With the help of metal molds, the ends were cast in epoxy resin. Figure 8 shows a sample ready for testing.

To determine the tensile strength of rovings embedded in TF10 CARBOrefit^©^ fine concrete, six tensile specimens with three profiled yarns each and a concrete cover of 5 mm were produced for each yarn series by laminating. This was done in a formwork in which the individual yarns were fixed and aligned with a yarn spacing of 13 mm. Then, the fine-grained concrete was filled in; first, a bottom layer (Figure 9) was subsequently slightly compressed. The top concrete layer was filled in and smoothed in a second step. The 120 cm long, 1 cm thick, and 33 cm wide plate was then covered with damp cloths. The plate was stored in water from the 2nd to the 7th day. From day 8 to day 28, in a climate chamber. Before the tensile tests, the plate was sawn in 5.2 cm wide stripes containing three yarns each. 

Specimens for the YPO tests were made by embedding single profiled carbon rovings as well as rovings with no profile and warp knitted rovings (reference) in the self-compacting fine-grained concrete BMK 45-220-2 in a cube formwork (Figure 10). One specimen consisted of two centered concrete blocks at the yarn ends and a free yarn segment of 120 mm in between the blocks. This was a clearly defined area in which composite failure could occur. The specimens were stored for seven days underwater and stored for additional 21 days in a climate chamber (20 °C and 65% relative humidity).

## 4. Results and Discussion

### 4.1. Processing Quality of Profiled Rovings 

In order to evaluate the processing quality, especially the filament arrangement of the profiled rovings, microsection analyses were conducted on different roving sections (Figure 11). The microscopic tests were performed at ITM’s textile–physical testing laboratory. At least five cross- and longitudinal sections of profiled rovings were analyzed. The microsections showed almost no air gaps between the filaments or polymer accumulations. From this, it is deduced that a very dense filament arrangement was achieved (Figure 11a). The longitudinal section (Figure 11b) visualized the filament course along the roving axis, where the filaments showed no apparent damage or deviation from the linear orientation between the profile sections. In the profile sections, the filaments showed a dense arrangement, which resembled the cross-section analysis. The dark spots in the longitudinal section (Figure 11b) are surface irregularities (filament detachment) due to preparation (cut of the roving) which caused the light to reflect away. 

In conclusion of the microscopic test series, the dense filament arrangement increased the inner bond of the impregnated roving and, according to Hahn et al., resulted in higher material utilization because almost no air gaps or polymer accumulations disturbed the load transmission between the single filaments [15]. The mechanical characterization of the tensile properties of the profiled rovings themselves, embedded rovings in concrete, as well as their bond behavior is discussed in Section 4.2, Section 4.3 and Section 4.4.

### 4.2. Tensile Strength of Single Rovings

The following diagrams illustrate the mean values of the tensile tests of the different series of fiber strand configurations with their standard deviation. For each series, at least seven to ten single specimens were tested according to DIN EN ISO 10618 [44]. An important part of the study was a comparison between tensile properties of dry rovings (Series 0, compare Table 4), impregnated rovings from a reference textile (Series R), impregnated rovings with no profile (Series 0T/0L), and impregnated rovings with a defined tetrahedral profile. This was to conclude the influence of the profile on tensile properties of the roving (Series 3/4). Hereby, the determined tensile strength (N/mm²) refers in all tests (dry and consolidated rovings) to the measured force (in N) divided by the dry and compact filament area of 1.81 mm². The composite dimensions of the impregnated rovings were neglected for the calculation of the tensile strength and Young’s Modulus because only the filaments transmit the load. The Young’s Modulus is the quotient of the absolute tensile strength difference and total elongation in the range of 0.15% and 0.9% elongation.

The diagrams in Figure 12 show the determined tensile strength and Young’s Modulus of dry CF-rovings (Series 0) in comparison to impregnated rovings with two different impregnation agents (Lefasol—Series 0L, Tecosit—Series 0T). The single standard deviation is specified with error bars.

As expected, tensile strength significantly increased by a factor of two through the impregnation of the rovings in contrast to dry rovings. The high tensile strength indicated a good and even distributed polymeric impregnation of the rovings resulting in an improved internal bond and load transmission between the filaments across the roving cross-section. A further influence was the different test setups. The dry rovings were tested according to ISO 3341 [36] with wrap specimen holders, as described in [35], whereas the impregnated rovings were tested with resinated clamping areas in metal clamps (see Section 3.4), allowing a better load introduction into the roving. The theoretical tensile strength of the carbon roving with 4300 MPa according to the datasheet [37] (density of 1.77 g/cm³) could not be achieved due to the inhomogeneous load introduction of the dry roving. The slightly different composition of the impregnation agents caused a little difference in achievable tensile properties. The tensile strength of impregnated rovings with Tecosit (Series 0T) was about 5% higher compared to impregnated rovings with Lefasol (Series 0L). Young’s Modulus of both impregnations was almost identical at around 235 GPa. Because of the better performance of impregnated, unprofiled rovings with Tecosit impregnation in contrast to Lefasol impregnated rovings, the following diagrams for the evaluation of the tensile properties of profiled rovings will only show results from profiled rovings with the impregnation agent Tecosit.

The averaged tensile properties of at least seven to ten tetrahedral profiled rovings with different profile configurations (medium profile—Series 3, strong profile—Series 4) are shown in the diagrams in Figure 13 and compared to impregnated rovings with no profile (Series 0T) and single rovings from the reference textile SITgrid 040 (Series R). All tested specimens were impregnated with Tecosit.

The tensile strength of the impregnated rovings without profile (Series 0T), the extracted rovings from the SITgrid 040 textile (Series R), and the middle profiled rovings (Series 3) were ~3400 MPa similarly high. The rovings with a stronger profile (Series 4) showed ~3200 MPa, a slight decrease in their tensile strength compared to the other variants. 

These results also mainly applied to the E-Modulus of the different rovings. The rovings with no profile achieved ~240 GPa, the highest due to the straight-oriented filaments. The extracted rovings from the grid-like non-crimp fabric SITgrid 040 showed ~230 GPa, a slightly reduced E-Modulus. The thesis is that the roving with no profile (Series 0T) was more uniformly impregnated, which resulted in a more uniform utilization of the single filaments. In addition, the roving was impregnated in a state of almost no tension, whereas the roving from the reference textile (Series R) was warp knitted before impregnation. The constriction of the roving with the knitting thread impeded a uniform impregnation [15]. The profiled rovings (Series 3/4) showed a slight reduction in Young’s Modulus to around 200 GPa. This was a logical consequence of profiling because the filaments were slightly deviated from an absolute straight orientation along the yarn axis, and therefore the applied load was not induced straight into the filaments, resulting in a slightly incomplete utilization of the anisotropic fiber properties. The strongly profiled rovings had the lowest tensile stiffness compared to the other profile configurations. Nonetheless, the tensile properties of the profiled rovings were very high due to the almost uniform reorientation of all the filaments during the shaping process (see Section 4.1). 

In conclusion, the profiled rovings achieved a high tensile strength of over 3000 MPa in addition to a high tensile stiffness of around 200 GPa.

### 4.3. Tensile Strength of Concrete Embedded Rovings 

Results from tensile tests with embedded carbon fiber heavy tows (CFHT) (impregnation: Lefasol, compare Table 4) are shown. These yarns were profiled in the prototype unit (Series 2_P). Of the total of six samples produced, only five could be included in the evaluation, as the DD1 had slipped in the first attempt. Figure 14 shows the single yarn tension–elongation curves of individual test specimens (each specimen numbered 1 to 5) with a very low scatter and the averaged mean curve (red) of all tested specimens. 

A mean yarn tensile stress of 3382 MPa (refers to a dry and compacted filament area of 1.81 mm²) was determined, which is right in the range of the failure strength in the yarn tensile tests (see Section 4.2). This was further confirmation that the profiling does not cause any process-related damage to the carbon fibers. Furthermore, it can be stated that the two experimental methods (tensile test of single yarns and tensile test of embedded yarns) are well suitable to determine the tensile strength of carbon rovings.

Figure 15 shows fractured test specimens of concrete embedded rovings after testing.

It is visible, that yarn rupture occurred in all tests, indicating that the bond was sufficient enough to transfer the full load from the concrete matrix into the rovings. The profiling of the yarns was still clearly visible after the tensile test. There was no pull-out of the yarns from the areas of load application. At the moment of failure, a complete spalling of the concrete occurred in the measuring area. No splitting or delamination cracks could be observed during the tests.

Yarns profiled in the laboratory unit were also tested. Figure 16 shows the averaged mean value yarn tensile stress–strain curves of six samples from Series 2_P (prototype unit, Lefasol), Series 1 (laboratory unit, Lefasol), and Series 3 (laboratory unit, Tecosit). All values were in the order of magnitude of the single yarn tension test and referred to the compact filament area of 1.81 mm²; the results in the laboratory and the prototype system were very similar. The failure in each case was yarn breakage. The scatter was again very moderate, which means that the tensile properties are reproducible. The failure strength of the Lefasol impregnated yarns (Series 1) was slightly lower than that of Tecosit yarns (Series 3), which agrees with the results in Section 4.2. The varying slope of the curves indicates differences in yarn stiffness. This and the formation of cracks were not systematically investigated during these initial tests and therefore not discussed in detail here. Such considerations are the subject of ongoing and planned research.

### 4.4. Bond Behavior of Concrete Embedded Rovings

In order to compare the bond behavior of the rovings with different profiles as well as varied impregnation and consolidation parameters, pull-out tests were carried out at the Institute of Construction Materials (IfB) of the TU, Dresden, according to the described test setup (Section 3.3). On average, four test specimens per configuration were tested. 

The following diagrams show the averaged specific pull-out load– slip-deformation curves and the bond strength of the rovings, which was equal to the maximum pull-out load. Hereby, the specific pull-out load, and thus bond strength (in N/mm), refers to the measured bond force (in N) divided by the bond length. Because no tested specimen showed shear cone failure or partial separation of concrete from the sample surfaced, the initial bond length was a constant 50 mm.

The diagrams in Figure 17 illustrate the bond strength (b) and the measured curves (a) of continuously produced profiled rovings with long consolidation (Series 4_10 min) as well as of profiled rovings from the discontinuously working prototype unit (Series 4_P). Both were compared to impregnated rovings without profile (Series 0T) and fiber strands extracted from the reference textile SITgrid 040 (Series R).

Hereby, the profiled rovings achieved their highest bond strength at ~100 N/mm, which was up to 40% higher compared to the reference (~70 N/mm) and almost five times the bond strength of the straight roving (~20 N/mm). The pull-out curves proved that by the defined profiling of the roving, the transmittable bond force can be increased and controlled, depending on the profile configuration.

As can be seen in diagram (a), all processed rovings initially showed approximately the same level of bond stiffness. However, the reference rovings (Series R) did not have a distinct profile geometry except for the oscillating geometry of the oval cross-section due to the warp knitting process, which caused a lower form-fit effect compared to the profiled rovings (Series 4). Therefore, the bond force was lower than for profiled rovings. As expected, the straight rovings without profile (Series 0T) showed the lowest bond performance (~20 N/mm). Here, the bond mechanism was mainly of a chemical nature due to adhesion between the roving surface and the surrounding concrete matrix.

During tests, all profiled rovings showed yarn rupture; thus, only incomplete pull-out curves were recorded. It indicated that the bond length of 50 mm is sufficient for a complete transfer of the tensile load inside the yarn to the concrete matrix via bond forces. Straight rovings and yarns from the reference textile (Ref.), were pulled out completely. Thus, the bond strength could be measured. Therefore, the strongly profiled rovings showed a much better bond performance compared to non-profiled reinforcement yarns.

The profiled rovings from the prototype unit (Series 4_P) achieved ~110 N/mm, the highest bond strength, as well as a higher bond stiffness compared to the strongly profiled rovings from the laboratory unit (Series 4_10 min). One reason is seen in the different profiling processes. Profiling in the prototype unit was a static process. Constant high pressure was applied—shaping and compaction of the roving were very uniform (see Section 2). It is believed that the profile produced in this way offers greater resistance to deformation under load than the yarns shaped in the laboratory plant. Here, the profiling tools were moved, and at the current stage of development, a truly constant pressure could not yet be guaranteed. The targeted control of the process parameters is subject to development to date.

Figure 18 illustrates bond behavior (a) and bond strength (b) of continuously produced profiled rovings with different profile configurations (no profile—Series 0T, medium profile—Series 3, and strong profile—Series 4). In comparison to rovings with no profile (~20 N/mm), medium-profiled rovings reached ~40 N/mm, about twice the bond strength. The stronger profiled rovings achieved ca. 80 N/mm, four times the bond force compared to rovings with no profile. Furthermore, profiled rovings showed a much higher bond stiffness indicated by the steeper increase at the beginning of the pull-out load–slip deformation curve. Hereby, especially the profiled rovings showed a continuous increase in the bond force and a distinct plateau at the maximum bond force, which indicated a stable mechanical interlock of the roving and the surrounding concrete matrix. 

Figure 19a shows the bond behavior of tetrahedral profiled rovings with strong profile (Series 4) in dependence on the solid contents of the impregnation Tecosit. It is evident that a reduced solid content of the polymeric dispersion led to a decrease in bond strength and stiffness. Furthermore, the reduction in solid content from 50% to 30% resulted in a decrease in transmittable bond loads of about 40% (from ~80 N/mm to ~50 N/mm). A possible reason is a smaller resistance against deformation of the profile during pull-out due to reduced impregnation content. The hypothesis is that the profile of the roving deformed differently under stress (depending on impregnation and consolidation parameters), resulting in a less corrugated profile and hence, reduced bond properties when the solid content of the impregnation is reduced.

The influence of consolidation time on rovings with the same profile and solid content is illustrated in Figure 19b. Herby, a longer-lasting consolidation by a heat input with IR radiation for 10 min (IR-specification see Section 3.1) resulted in a significantly higher maximum bond strength as well as a higher bond stiffness (steeper increase in the curve). According to the previously formulated thesis, an intensified consolidation of the roving resulted in higher resistance against deformation of the profile. It therefore allowed the transmission of higher bond forces before failure or deformation of the rovings. The thesis will be tested in further research studies via an optical analysis of the roving geometry during tensile tests with a high-speed camera, which allows detecting small deformations or changes in the profile during the applied tensile stress.

## 5. Conclusions and Outlook 

In summary, the profiled carbon rovings transmitted higher pull-out loads compared to unprofiled rovings and rovings extracted from warp knitted textiles. Due to the gentle shaping process and the good penetration of the impregnating agent, the rovings with a tetrahedral shape had high tensile strengths. 

It can be stated that:Better dense filament arrangement and better material utilization can be achieved by good penetration of impregnation agent and immediately following shaping of rovings (see Section 4.1)The developed shaping process created profiled rovings with a defined tetrahedral geometry that showed almost no decrease in their tensile properties (≤ 10%) compared to impregnated rovings with no profile (see Figure 13).Tetrahedral-shaped rovings showed up to 500% the concrete bond strength compared to rovings with no profile (see Figure 17) and 140% of warp knitted rovings (that showed a slight waviness and roving constriction).Bond strength and bond stiffness depended on the profile geometry, as well as impregnation and consolidation parameters (see Figure 18 and Figure 19); a defined variation of the stated parameters enabled a modification of the bond behavior.A strong profile in combination with an intensive (long) consolidation and a high solid content of the impregnation agent (50%) resulted in a higher bond performance (see Figure 19) with a maximum bond strength of about 100 N/mm.

Based on the results, the tetrahedral shaping process of impregnated rovings shows a high potential to create high-performance textile reinforcement structures with concrete bond-optimized behavior and high tensile properties. Primarily, the complete form-fit-based anchoring between profiled roving and surrounding concrete matrix resulted in an increase in the maximum transmittable bond strength and high bond stiffness with up to five times the values of consolidated straight rovings. The reduction in the required bond length enabled a better material efficiency, especially considering the energy-intensive production of carbon fibers and reinforcements.

In summary, it can be stated that we see further potential in the profiling process to maximize bond strength and bond stiffness by optimization of the impregnation (yarn spreading for even impregnation distribution) and consolidation process (focused energy input for intensified consolidation). Additionally, the investigation of influential parameters on bond behavior and their targeted adjustment could enable a predictable design of CRC structures with specific and application-oriented bond behavior. Therefore, extensive basic research is planned to find a method for the targeted adjustment of strength and composite properties through the defined and variable profiling of carbon yarns in addition to a numerical description of the bond behavior. 

Carbon fiber reinforcements with specifically adjustable properties will clearly increase the material efficiency of carbon-reinforced concrete in the future in the areas of new construction and strengthening. In the context of new construction, we are thinking of currently researched, material-efficient structural elements [49]. In the case of component strengthening, for example, shortened end anchorage and overlap lengths will improve handling. Additionally, the lower material consumption reduces costs, which increases competitiveness compared to other reinforcement methods.

## Figures and Tables

**Figure 1 materials-15-05581-f001:**
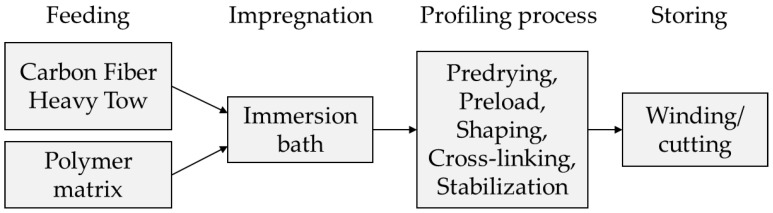
Schematic diagram of the profiling process.

**Figure 2 materials-15-05581-f002:**
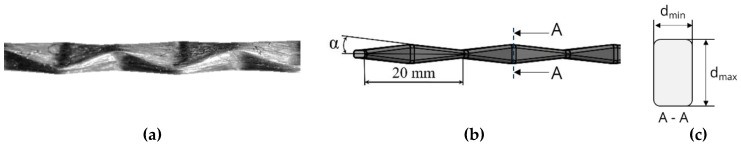
Profiled carbon roving with a tetrahedral geometry: (**a**) photography; (**b**) schematic illustration; (**c**) schematic cross-section.

**Figure 3 materials-15-05581-f003:**
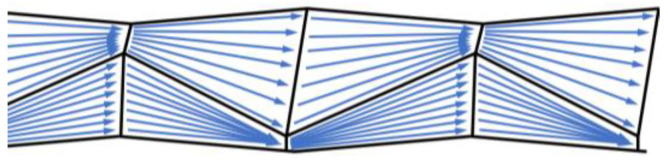
Schematic filament orientation of tetrahedral profiled rovings.

**Figure 4 materials-15-05581-f004:**
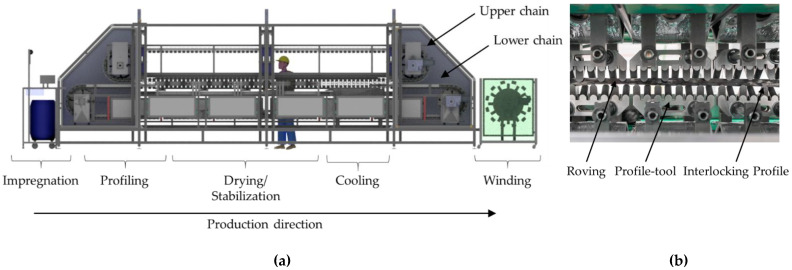
Function of the laboratory profiling unit: (**a**) schematic illustration; (**b**) interlocking profiling tools (**b**).

**Figure 5 materials-15-05581-f005:**
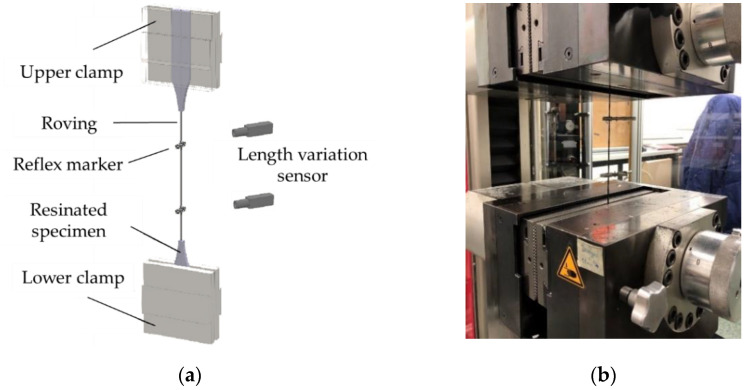
Test setup for yarn tensile test on basis of DIN EN ISO 10618 at ITM: (**a**) clamping principle; (**b**) test stand [35].

**Figure 6 materials-15-05581-f006:**
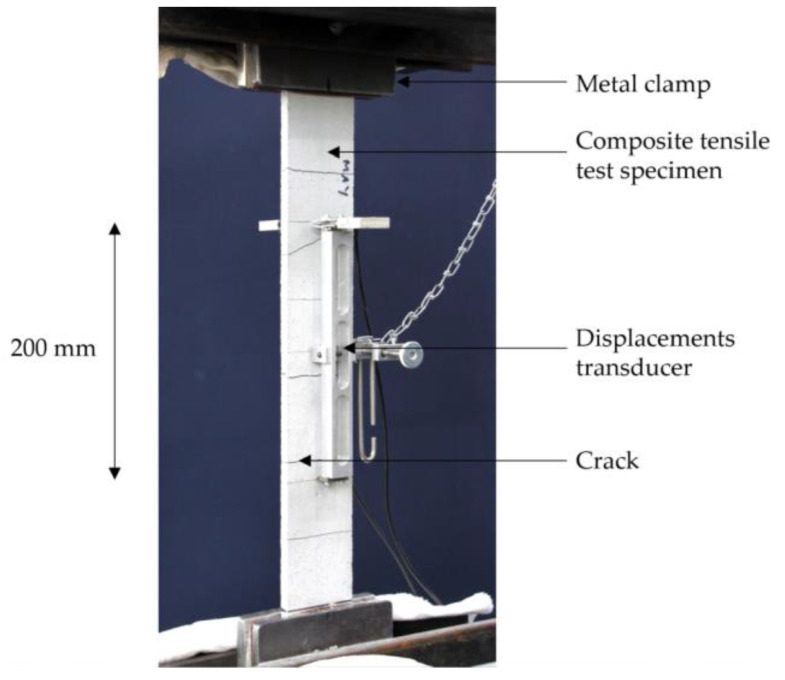
Characterization of carbon-reinforced concrete samples; test setup and measurement equipment for tensile tests on the composite.

**Figure 7 materials-15-05581-f007:**
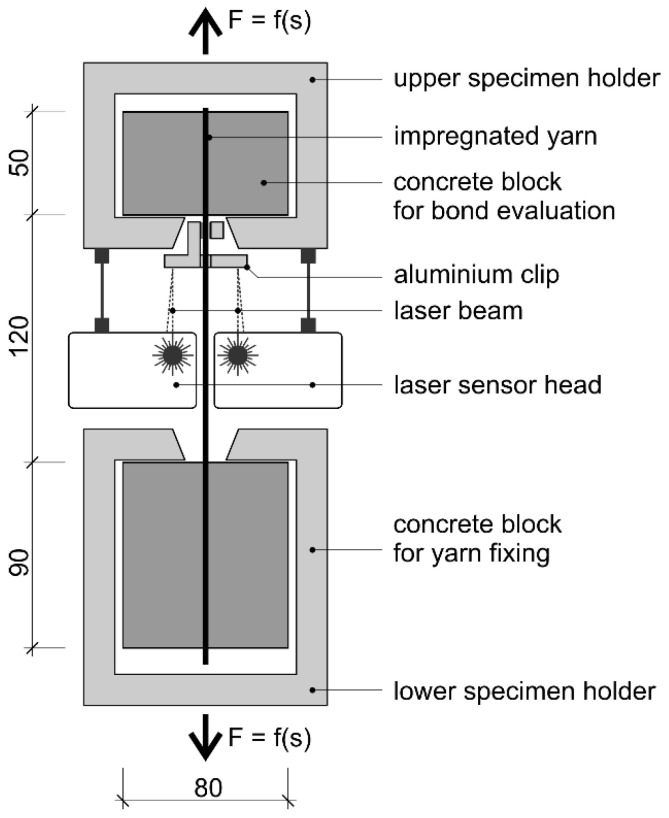
Schematic test-setup for single yarn pull-out (YPO) test (dimensions in mm) [27,47,48].

**Figure 8 materials-15-05581-f008:**
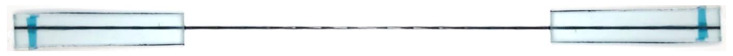
Tensile test specimen of a profiled roving with EP-resinated ends.

**Figure 9 materials-15-05581-f009:**
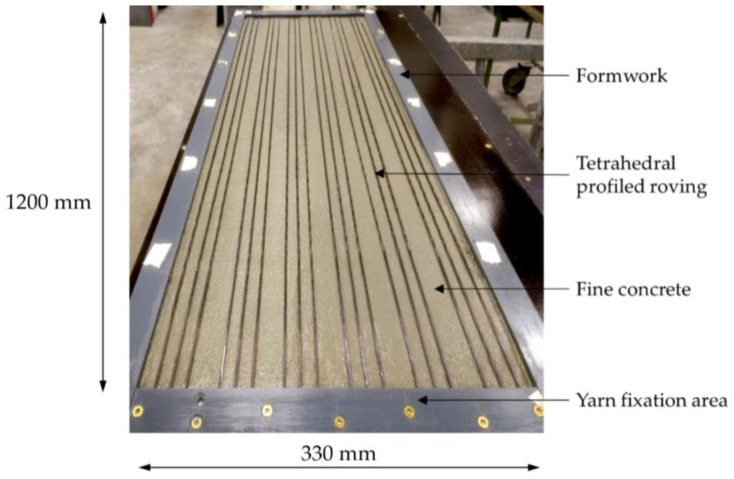
Laminating of tensile specimen: first concrete layer and straight yarns (three per sample).

**Figure 10 materials-15-05581-f010:**
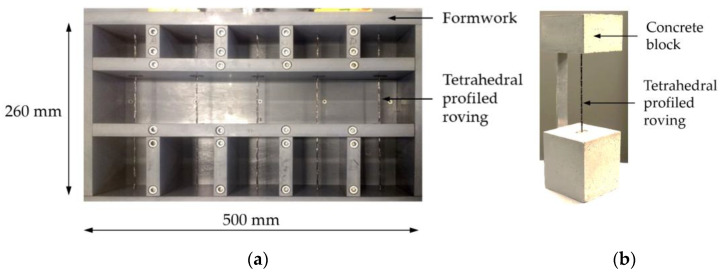
YPO specimen: (**a**) formwork with single yarns; (**b**) test specimen ready for pull-out.

**Figure 11 materials-15-05581-f011:**
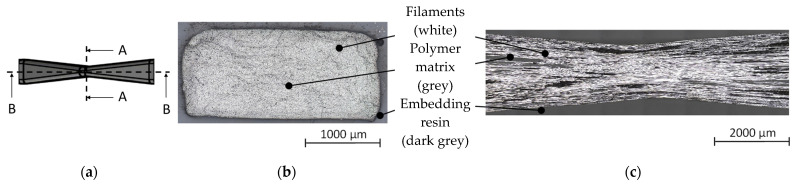
Exemplary microsection of profiled rovings: (**a**) schematic illustration of the microsection directions; (**b**) cross-section (A–A); (**c**) longitudinal section along the roving axis (B–B).

**Figure 12 materials-15-05581-f012:**
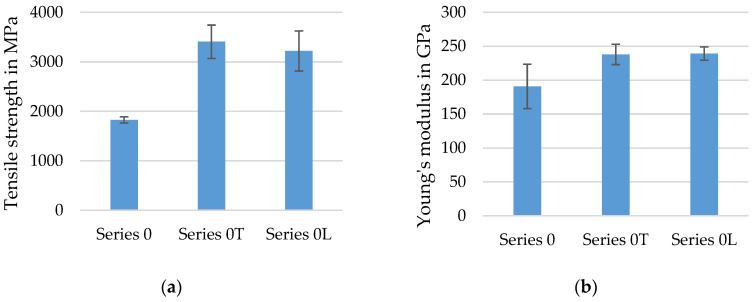
Tensile strength (**a**) and Young’s Modulus (**b**) of dry and impregnated CF-rovings.

**Figure 13 materials-15-05581-f013:**
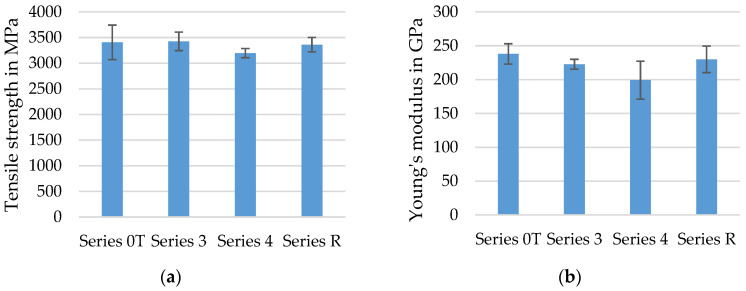
Tensile strength (**a**) and Young’s Modulus (**b**) of CF-rovings with different profiles.

**Figure 14 materials-15-05581-f014:**
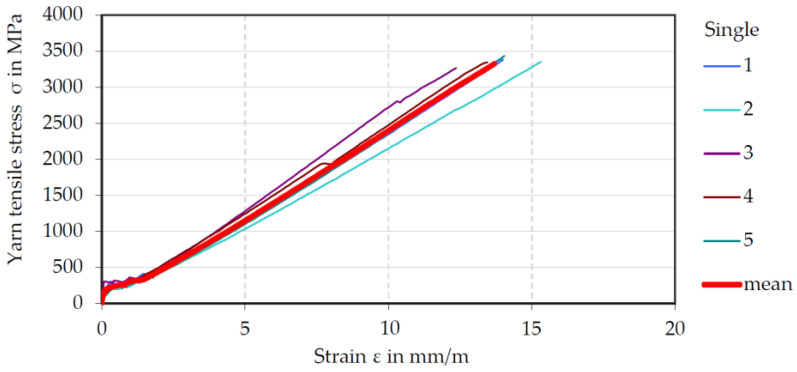
Stress–strain curves of CF-rovings profiled in the prototype unit. 1: specimen 1; 2: specimen 2; 3: specimen 3; 4: specimen 4; 5: specimen 5, mean: average of all tested specimens.

**Figure 15 materials-15-05581-f015:**
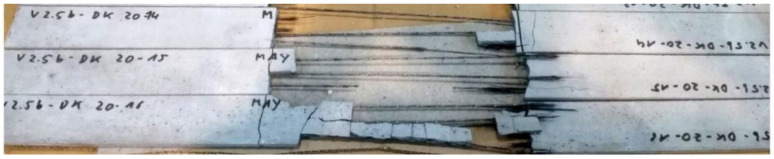
Tensile test specimen of concrete embedded rovings after failure.

**Figure 16 materials-15-05581-f016:**
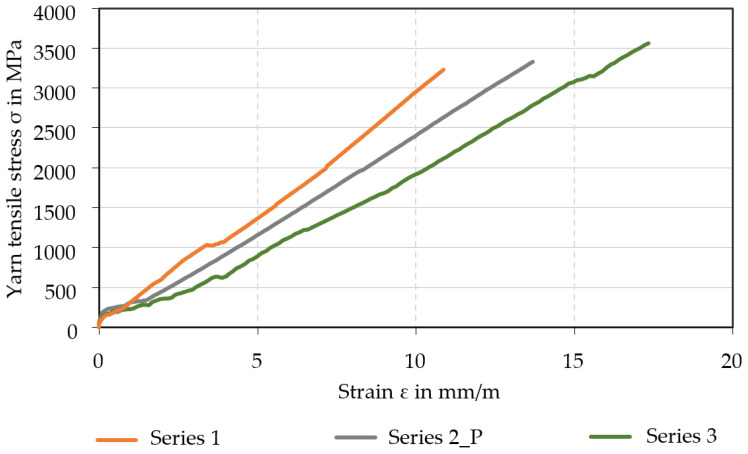
Averaged stress–strain curves of different profiled CF-rovings.

**Figure 17 materials-15-05581-f017:**
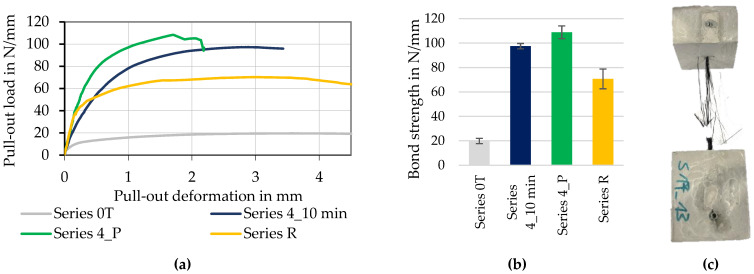
Specific pull-out load–slip deformation curves (**a**) and bond strength (**b**) of CF-rovings with different profile configurations and a specimen for the bond tests after failure (**c**).

**Figure 18 materials-15-05581-f018:**
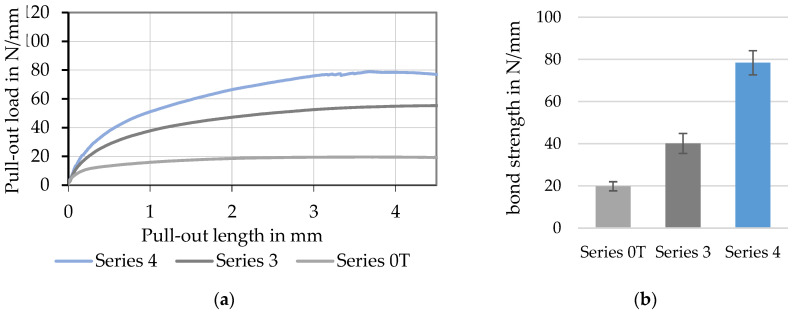
Specific pull-out load–slip deformation curves (**a**) and bond strength (**b**) of tetrahedral profiled CF-rovings with different profile configurations.

**Figure 19 materials-15-05581-f019:**
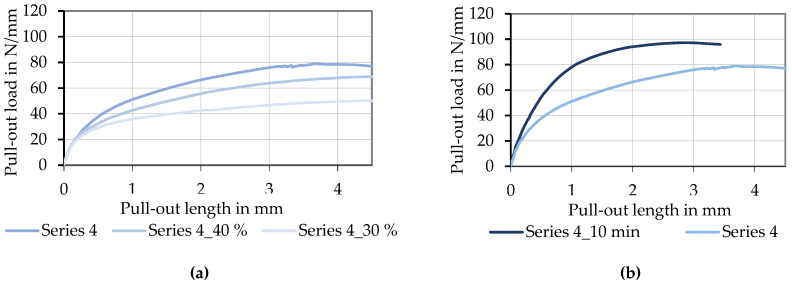
Specific pull-out load–slip deformation curves of tetrahedral profiled CF-rovings (strong profile) with different solid contents of Tecosit (**a**) and different consolidation times (**b**).

**Table 1 materials-15-05581-t001:** Properties of the dry yarn.

Property	Description/Values ^a^
Fiber material	Teijin Tenax-E STS 40 F13 48K 3200 tex Carbon roving
Density in g/cm³	1.77
Fineness in tex	3215
Tensile strength in MPa	1827 ^b^
Elastic modulus in GPa	188
Ultimate strain in %	1.20

^a^ Data according to the manufacturer’s specifications, unless otherwise stated [37]. ^b^ Determined in single yarn tensile tests at the ITM acc. to ISO 3341 [36].

**Table 2 materials-15-05581-t002:** Properties of the impregnation agents, data according to manufacturer’s specifications (Lefasol: [38], Tecosit: [39]).

Impregnation Agent				
Product Name	Characteristics	Base- Material	Solid Content in %	Linking Temperature in °C
TECOSIT CC 1000 (CHT Germany GmbH)	Aqueous polymer dispersion	Polyacrylate	47 ± 1	160
Lefasol BT 91001-1 (Lefatex Chemie GmbH)		Polystyrol	52 ± 1.5	150–160

**Table 3 materials-15-05581-t003:** Profile characteristics of the different rovings.

Roving
Configuration	Geometry	Dimension (~)	Cross-Section	Illustration
Without defined profile
Dry yarn	Band-shaped	Variable (no internal bond)		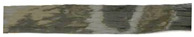
Impregnated roving	Circular	d = 2 mm	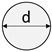	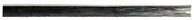
Roving from textile	Elliptical	d_1_ = 3.3 mmd_2_ = 1.3 mm	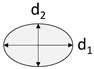	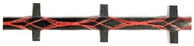
With defined profile
Tetrahedral profiled roving	Medium profile	d_diff_ = 0.6 mmα = 3°	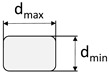	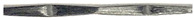
Strong profile	d_diff_ = 1.0 mmα = 5°	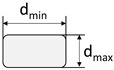	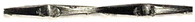

**Table 4 materials-15-05581-t004:** Properties of different rovings; Index “P” for prototype unit.

Roving Configuration	Sample	Parameter				
		Roving Geometry	Profile Unit	Impreg- Nation Material	Solid Content in %	Consoli- Dation Time in Min
Rovings without profile						
Dry yarn	Series 0	-	-	-	-	-
Impregnated roving	Series 0L	Circular		Lefasol	50	4
	Series 0T			Tecosit		
Roving from textile (Ref.)	Series R	Elliptical				unknown
Profiled rovings from prototype unit						
Profiled roving	Series 2_P	Tetrahedral Strong	Prototype unit	Lefasol	50	4
	Series 4_P			Tecosit		
Profiled rovings from laboratory unit with different profiles and impregnation agents						
Profiled roving	Series 1	Tetrahedral Medium	Laboratory unit	Lefasol	50	4
	Series 2	Tetrahedral Strong				
	Series 3	Tetrahedral Medium		Tecosit		
	Series 4	Tetrahedral Strong				
Profiled rovings from laboratory unit with different solid content and consolidation						
Profiled roving	Series 4_30%	Tetrahedral Strong	Laboratory unit	Tecosit	30	4
	Series 4_40%				40	
	Series 4_10 min				50	10

**Table 5 materials-15-05581-t005:** Concrete properties, minimum values after 28 days.

Concrete Property	TF 10 CARBOrefit^®^ Fine Concrete [42]	BMK-45-220-2
Compressive strength in MPa	≥80	≥105
Bending tensile strength in MPa	≥6	≥11.5
Maximum grain size in mm	1	2

## Data Availability

Data is contained within the article.

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
