# Peer review of "Bond Modification of Carbon Rovings through Profiling"

_materials, 2022, doi:10.3390/ma15165581_

Round 1

Reviewer 1 Report

The article focuses on the influence of yarn profiling on selected mechanical properties. It is perfectly presented, achieves excellent results. Especially single yarn pull-out test clearly presents the influence of yarn profiling, as well as the impact of yarn impregnation itself. I would only have small comments and questions to the next explanation:

1) I did not find how many samples were created and tested for each groups.

2) For the test set up for single yarn pull-out test, concrete block for bond evaluation, embedment length of 50 mm. I do not see measures against shear cone. During the pull-out process, part of the concrete may be separated at the sample surface, the considered embedment lenghlt is than not accurate. It is described for example in ACI 440.3R-03 for FRP specimens, where part of the reinforcement is separated from the concrete, so the shear cone does not arise. Was this phenomenon somehow considered in the experiment?

3) Line 379, Young's modulus: Can you please specify the calculation procedure?

4)For the results of the tensile strength (page 12 of 21), the Series 0T and Series 0l - tensile strength is almost twice as high as the theoretical maximum value of 1827 MPa (yarn) with considered area of the yarn to the calculation of 1.81 mm2. This shows perfect homogenication of yarn. But does the polymer matrix really have such a major impact on the tensile strength of composite? And further, series 0, without impregnation, with logically weak interaction between single fibrils (filaments). Did 100 % theoretical bearing capacity really achieved according to the technical data sheete of yarn? How is it possible without any impregnation? Please comment.

5) Figure 14, 15: Are these specific curves or average curves? Please add as you have with other pictures.

Author Response

Dear Sir or Madam,

thank you for your helpfull comments. I have revised the manuscript according to your and other reviewers suggestions.

1) the number/averaged number of tested specimens was added for each test series

2) There was no failure of the shear cone due to relatively small yarn cross sections in contrast to rebars. It has been added, that no concrete parts seperated an therefore the bond length is a constant of 50 mm.

3) I added the calculation procedure at the concerning line.

4) At the corresponding section I added, that the dry yarn and impregnated yarn were tested differently, the impregnation improves the load transmission and the test set up for dry yarns doesn`t allouw a even load introduction therfore resulting in a reduced bond strength compared to the data sheet. The composite dimensions were negleted, becaus only the filaments transmit the load and the polymer increases fiber friction and load transmission.

5) I added the indication of averaged curves

If my changes do not suffice or satisfy you, please let me know.

Kind regards

Paul Penzel

Reviewer 2 Report

The conducted research is very interesting and useful, and the research work is also abundant. The manuscript is well written, the reviewer gives some minor suggestions:

1. Try to improve the title of the paper, it is too short and does not fully reflect the research point.

2. The organization of Abstract is not well done. Much of the text is devoted to the background. And it is hard to distinguish research aim, methods and main conclusions. The authors are suggested to prepare the Abstract according to a regular form, basic elements including background, aim, methods and conclusions are needed.

3. The statement in Introduction is a little scattered, it is suggested to be summarized closer to what will be done in the following research.

4. Can the authors add a partial schematic to Figure 4 to show how to realize the tetrahedral structure of rovings?

5. Please check the writing format of the text. The citation format of some references is inappropriate, such as “Exemplary for overview publications, [1–6] are mentioned.”, a direct statement of what has been done in these papers will be better. The Figure 2 appears before Figure 1. The authors who contribute to a certain Figure were appended to the figure title, such as (Source: P. Penzel), it's unnecessary to do that and will lead to misunderstandings, it seems these Figures from other researchers’ output. There are many similar problems.

6. Please mark the corresponding dimensions in the Figure 3, Figure 6, Figure 9 and Figure 10.

7. Figure 14 is not well displayed, and more explanation about the results is suggested, such as the characteristics of tensile strength, strain and fracture of samples.

8. Try to short the conclusions and outlook, there are too many tips, just need to give out the most important conclusions and ideas.

Author Response

Dear Sir or Madam,

thank you for your helpfull comments. I have revised the manuscript according to your and other reviewers suggestions.

  1. Title: Yarn Modification through Profiling --> Bond Modification of Carbon Rovings through Profiling
  2. The abstract is revised; aim, methods and conclusios are emphasized
  3. the statements in the introduction have been added acc. to the following research
  4. a figure of the interlocking profiling tools for creating the tetrahedral shape has been added
  5. A direct statement has been added; in my verions is the appearence of the figures correct - please let me know if not so; the contributing authors of the figures have been deleted
  6. dimensions have been added in Fig. 6/9/10. Figure 3 is schematical and only for illustration. The dimensions are incorrect to the original profile and therefore not usefull
  7. Figure 14 has been split and reorganized. The characteristics are now more emphasized
  8. Conclusion and outlook have been shoretend and summarized

If my changes do not suffice or satisfy you, please let me know.

Kind regards

Paul Penzel

Reviewer 3 Report

COMMENTS

In many figures, the authors say: (Source: ....)  OR (Sources: ....  )

Have they drawn the figures by themselves OR  Did they get copyright permission?

5. Conclusions   [ NOT 5. Conclusions and outlook]

Author Response

Dear Sir or Madam,

thank you for your comments. I have revised the manuscript according to your and other reviewers suggestions.

If my changes do not suffice or satisfy you, please let me know.

Kind regards

Paul Penzel